# Association of Heart Rate Variability with Obstructive Sleep Apnea in Adults

**DOI:** 10.3390/medicina59030471

**Published:** 2023-02-27

**Authors:** Yen-Chang Lin, Jui-Kun Chiang, Chih-Ming Lu, Yee-Hsin Kao

**Affiliations:** 1Nature Dental Clinic, Puli Township, Nantou 404, Taiwan; 2Department of Family Medicine, Dalin Tzu Chi Hospital, Buddhist Tzu Chi Medical Foundation, No. 2, Minsheng Road, Dalin, Chiayi 622, Taiwan; 3Department of Urology, Dalin Tzu Chi Hospital, Buddhist Tzu Chi Medical Foundation, No. 2, Minsheng Road, Dalin, Chiayi 622, Taiwan; 4Department of Family Medicine, Tainan Municipal Hospital (Managed by Show Chwan Medical Care Corporation), 670 Chung-Te Road, Tainan 701, Taiwan

**Keywords:** heart rate variability (HRV), the apnea-hypopnea index (AHI), obstructive sleep apnea (OSA), polysomnography (PSG)

## Abstract

*Background and Objectives*: Heart rate variability (HRV) analysis is a noninvasive method used to examine autonomic system function, and the clinical applications of HRV analysis have been well documented. The aim of this study is to investigate the association between HRV and the apnea–hypopnea index (AHI) in patients referred for polysomnography (PSG) for obstructive sleep apnea (OSA) diagnosis. *Materials and Methods*: Patients underwent whole-night PSG. Data on nocturnal HRV and AHI were analyzed. We determined the correlation of time- and frequency-domain parameters of HRV with the AHI. *Results*: A total of 62 participants (50 men and 12 women) were enrolled. The mean age, body mass index (BMI), neck circumference, and AHI score of the patients were 44.4 ± 11.5 years, 28.7 ± 5.2, 40.2 ± 4.8 cm, and 32.1 ± 27.0, respectively. The log root mean square of successive differences between normal heartbeats (RMSSD) were negatively correlated with BMI (*p* = 0.034) and neck circumference (*p* = 0.003). The log absolute power of the low-frequency band over high-frequency band (LF/HF) ratio was positively correlated with the AHI (*p* = 0.006). A higher log LF/HF power ratio (*β* = 5.01, *p* = 0.029) and BMI (*β* = 2.20, *p* < 0.001) were associated with a higher AHI value in multiple linear regression analysis. *Conclusions*: A higher log LF/HF power ratio and BMI were positively and significantly associated with the AHI during whole-night PSG in adult patients.

## 1. Introduction

Sleep occupies almost one-third of our lives. The American Academy of Sleep Medicine and the Sleep Research Society recommend ≥7 h of sleep per night for adults to ensure optimal health [1]. Sleep is a highly complex phenomenon regulated partly by the autonomic nervous system. Both short and long sleep durations and sleep disorders, mainly OSA, adversely affect cardiovascular and metabolic disorders [2]. OSA is a severe sleep disorder that may deteriorate the quality of life and lead to hypertension and cardiovascular and cerebrovascular diseases [3]. The OSA severity is evaluated based on the AHI determined through PSG. AHI, by definition, is the sum of the numbers of “apnea” and “hypopnea” events per hour during sleep. As for “apnea” and “hypopnea”, respectively, the former is delimited by cessation of airflow for at least 10 s, while the latter by reduction in airflow by at least 30% for at least 10 s with decrease in oxygen saturation. AHI values of 5–15, 15–30, and >30 with associated symptoms (e.g., excessive daytime sleepiness, fatigue, impaired cognition, or a spouse’s report of disruptive snoring) indicate mild, moderate, and severe OSA, respectively [4]; or OSA is defined as an AHI ≥ 15, regardless of the associated symptoms [5]. Meta-analyses have reported that OSA is associated with diabetes mellitus [6], stroke [7], total cardiovascular diseases [7], and all-cause mortality [8].

Autonomic dysfunction is associated with various pathological conditions [9], including hyperglycemia, high blood pressure, high triglyceride levels, low high-density lipoprotein cholesterol levels, high BMI, incident diabetes, cardiovascular disease (CVD), and high mortality [10,11,12]. Three tests, namely the RR variation, Valsalva maneuver, and postural blood pressure testing, have been recommended for the longitudinal testing of the cardiovascular autonomic system since 1992 [13]. HRV analysis is a noninvasive method to examine autonomic system function.

One meta-analysis reported that lower HRV was associated with a 46% higher risk of cardiovascular events, which was significant for patients with acute myocardial infarction, and a 112% higher risk of all-cause death [14]. Another meta-analysis demonstrated that lower HRV was associated with a 32–45% increased risk of a first cardiovascular event in individuals without known CVD [15]. Studies have reported the association of HRV with higher hemoglobin A1C (HbA1c) levels in young adults with diabetes and patients with diabetic autonomic neuropathy [16].

HRV analysis has been the most commonly used to monitor autonomic changes during sleep, and HRV was associated with OSA [17,18]. Ischemic cardiovascular events in OSA have multifactorial etiology, including high sympathetic activity, endothelial dysfunction, inflammation, and oxidative stress [18,19,20,21], with the primary contributor being sympathetic overactivity [22]. The sympathetic activity was synchronized with the repetitive episodes of apnea occurring continuously throughout the sleep of patients with OSA [21]. One study reported that using HRV analysis in OSA provides insights into cardiac autonomic control across sleep stages [18]. Another systemic review reported that adults with OSA had higher sympathetic components and lower parasympathetic tone [23]. However, the parasympathetic tone, RMSSD, was not significantly lower in adults with OSA in some previous studies [23]. Because snoring and sleep apnea were found during sleeping in the night, the association between HRV and OSA might be better investigated during sleep. The aim of this study was to investigate the association between HRV and OSA severity by considering frequency and time domains during whole-night PSG.

## 2. Materials and Methods

### 2.1. Ethical Considerations

The study protocol was reviewed and approved by the institutional review board of the Tainan Municipal Hospital (Managed by Show Chwan Medical Care Corporation) (SCMH_IRB No: 1090508) and the Research Ethics Committee of the Buddhist Dalin Tzu Chi Hospital, Taiwan (No. B10901020).

### 2.2. Materials

#### 2.2.1. Data Collection from PSG

Patients underwent PSG (EMBLA N7000 system, Embla Inc., Broomfield, CO, USA). We examined electrophysiological signals for heart activity analysis, pulse oximetry readings, and airflow by using nasal pressure and oronasal thermal sensors, body position, actigraphy data, and thoracic and abdominal movements [24].

#### 2.2.2. HRV Measurement

Time- and frequency-domain analyses were performed to evaluate HRV. The time-domain measurement indices included the standard deviation of normal-to-normal (NN) intervals (SDNN), the standard deviation of average NN intervals for each 5-min segment of an HRV recording (SDANN), percentage of successive RR intervals that differ by >50 ms (pNN50), baseline width of the RR interval histogram (TINN), the standard deviation of successive RR interval differences (SDSD), and the root mean square of successive differences (RMSSD). The RMSSD was converted by logarithmic transformation (log RMSSD), as suggested by Nakamura et al. (Nakamura et al., 2015). Both the RMSSD and log RMSSD are the recognized markers of parasympathetic activity [25].

Frequency-domain analysis was performed in a 1024 sample (8.5 min) window by using the fast Fourier transform applied to three overlapping 512 sample sub-windows within the 1024 coherence windows. For each time segment, the algorithm estimated the power spectral density. The spectrum resulted from the sampling and Hamming windows. The power spectrum was quantified by integrating it into frequency-domain indices with a high-frequency (HF) power (0.15–0.4 Hz), low-frequency (LF) power (0.04–0.15 Hz), and the ratio of LF power to HF power (log LF/HF ratio).

### 2.3. Method

#### 2.3.1. Data Design and Setting

We recruited 62 (50 men and 12 women) consecutive participants referred to the sleep unit of a southern teaching hospital in Taiwan for clinically suspected OSA after excluding 5 individuals who failed to meet the inclusion criteria of this current program. Most patients were referred from ENT and the internal medical department in the same hospital from July 2020 to June 2021. The inclusion criteria were patients who received PSG due to snoring or the severity of OSA. We excluded the subjects with severe cardiovascular disorders, severe neuromuscular disorders, previous surgery for snoring and sleep apnea, or taking medications that affect the sympathetic nervous system (e.g., beta-blockers, alpha-blockers, and centrally acting drugs). We obtained informed consent from all patients prior to their enrollment in the study.

#### 2.3.2. Study Outcome

Heart rate variability analyses including time- and frequency-domain measurements from PSG, and demographic factors were collected to evaluate the independent factors associated with the apnea–hypopnea index as the measures of OSA severity. Since the American Academy of Sleep Medicine has long established a threshold of 5 events of apnea/hypopnea per hour with OSA symptoms (unintentional sleep episodes during wakefulness; daytime sleepiness; unrefreshing sleep; fatigue; insomnia; waking up breath-holding, gasping, or choking; or the bed partner describing loud snoring, breathing interruptions, or both during the patient’s sleep) or 15 events per hour (with or without OSA symptoms) as criterion for OSA cases [4], we would set the cutoff of AHI ≥ 15 events/h as the criteria for the OSA cases in our current study regardless of the associated symptoms.

### 2.4. Statistical Analysis

The PSG could record several channels of data including the electroencephalogram, electrooculogram, and electrocardiogram. Electrocardiogram data were downloaded, Hilbert transformed and analyzed using R with the ebm, seewave, pracma and RHRV packages. Data files were visually inspected for artifacts, and corrections were made manually or by using the software if necessary. Some HRV indices were naturally logarithmically transformed to reduce the skewness of data distribution [26]. Pearson’s correlation (*r*) was calculated to determine the relationship between two continuous variables. The correlation was considered to be strong if *r* > 0.5, moderate if *r* = 0.3–0.5, and weak if *r* = 0.1–0.3 [27]. Multiple linear regression analysis was performed to analyze the association between HRV indices and the AHI after adjustment for covariates [28]. All factors listed in Table 1 and Table 2 were included during the regression analysis. Data entry and analysis were performed using the free R software, version 4.0.3 (R Foundation for Statistical Computing, Vienna, Australia). All statistical assessments were two-sided, and statistical significance was set at the 0.05 level.

## 3. Results

A total of 62 participants (50 men and 12 women) referred for PSG were enrolled in this study, and the mean recording time of PSG was 6.9 ± 0.3 h. The mean age, BMI, neck circumference, and AHI score of the patients were 44.4 ± 11.5 years, 28.7 ± 5.2, 40.2 ± 4.8 cm, and 32.1 ± 27.0, respectively. Thirty-nine patients (62.9%) were classified as OSA with AHI ≥ 15. Additionally, neck circumference, hypoxemia index and arousal index were all significantly higher in the AHI ≥ 15 group than the AHI < 15 group (41.2 ± 5.1 cm vs. 38.5 ± 3.8 cm, *p* = 0.016; 12.5 ± 9.7/hour vs. 5.0 ± 2.5/hour, *p* = 0.004; 25.2 ± 14.9/hour vs. 10.4 ± 3.8/hour, *p* < 0.001, respectively) (Table 1).

The HRV data handling and analysis is shown in Figure 1.

As shown in Table 2, the mean log SDNN was 4.8 ± 0.5 log (ms) and negatively correlated with BMI (*p* = 0.036) and neck circumference (*p* = 0.014).

The mean log RMSSD was 4.5 ± 0.6 log (ms) and negatively correlated with BMI (*p* = 0.034) and neck circumference (*p* = 0.003). The log LF/HF ratio was 0.1 ± 0.5 and was positively correlated with AHI (*p* = 0.006; Figure 2), neck circumference (*p* = 0.040) and hypoxemia index (*p* = 0.031) (Table 2).

The log RMSSD values were negatively correlated with AHI; however, this correlation was nonsignificant (*p* = 0.191; Figure 3).

Notably, AHI was positively correlated with arousal index (*r* = 0.70, *p* < 0.001) and hypoxemia index (*r* = 0.41, *p* = 0.001). The collinearity was also checked for variables selection. The results of the multiple linear regression analysis revealed that a higher log ratio of LF/HF power (*β* = 15.01, *p* = 0.029) and BMI (*β* = 2.20, *p* < 0.001) were associated with a higher AHI value (Table 3). The R^2^ value for this final model was 0.314. To better illustrate the effect of arousals and their associated sympathetic activation on heart rate variability, we further yield a model of the multiple linear regression analysis showing that only hypoxemia index (*β* = 0.019, *p* = 0.032) but not AHI (*β* = 0.005, *p* = 0.151) or arousal index (*β* = −0.003, *p* = 0.662) was significantly correlated with higher log ratio of LF/HF power.

## 4. Discussion

A higher log value indicates a higher original value. Accordingly, we observed that a higher log LF/HF power ratio and BMI were positively and significantly associated with the AHI in the adult patients who underwent whole-night PSG; that is, the patients with OSA had a higher log LF/HF power ratio during sleep. However, adults with OSA had no significantly lower parasympathetic tone, RMSSD.

HRV, which refers to fluctuations in time intervals between adjacent heartbeats, is an emergent property of interdependent regulatory systems that operates on different time scales to adapt to environmental and psychological challenges [29]. In this study, Ultra-Low Frequency (ULF), VLF, LF power, HF power, and the log LF/HF power ratio were examined in the frequency-domain analysis. Only the log LF/HF power ratio was significantly related to the AHI. The HF power component corresponded to respiratory sinus arrhythmia and is modulated only by the parasympathetic nervous system. The LF power component is jointly modulated by the sympathetic and parasympathetic nervous systems. The log LF/HF power ratio was determined to evaluate sympathovagal balance. A low log LF/HF power ratio reflects parasympathetic dominance, whereas a high log LF/HF power ratio indicates sympathetic dominance and low vagal activation [27]. A significantly decreased overall HRV exhibits a pattern of parasympathetic loss (lower RMSSD, PNN50, and HF power) with sympathetic overdrive (higher LF) and sympathovagal imbalance (higher log LF/HF power ratio) [16]. HRV can increase the risks of diabetes, obesity, osteoporosis, arthritis, Alzheimer’s disease, periodontal disease, cancer, frailty, and disability [9]. We observed that a higher log LF/HF power ratio was strongly associated with the AHI, which is an indicator of OSA. One study reported that a higher log LF/HF power ratio increased sympathetic tone and discordance in sympathovagal activity in patients with moderate OSA [17]. Thus, patients with OSA might have an increased risk of cardiovascular and cerebrovascular diseases.

The SDNN, SDANN, SDNNIDX, pNN50, TINN, SDSD, and RMSSD were included in the time-domain analysis. The RMSSD is the primary time-domain measure used to estimate vagally mediated changes reflected in HRV [30]. Compared with the pNN50, the RMSSD more favorably reflects changes in HRV [31]. One study reported that the RMSSD was measured to examine tonic vagal activity and was strongly correlated with HF power [32]. We analyzed the RMSSD and observed that it was negatively and significantly correlated with the log LF/HF power ratio (correlation: −0.633, *p* < 0.001) but was not significantly correlated with the AHI (correlation: −0.179, *p* = 0.234). Similar results were also found in the previous studies [23,33,34,35]. Other studies reported that RMSSD was lower in the OSA during night time, but it was not significantly lower in records from the daytime and 24 h recording [36]; and RMSSD was lower while patients with severe OSA. The explanations included different designs and sizes of samples. Another explanation might be the sympathetic components might be more sensitive than the parasympathetic components for adults with OSA. Yet, more studies should be warranted to demonstrate the rationale. In time-domain analyses, we noted that the log value of the SDNN was 4.8 ± 0.5. A study reported that a log SDNN value of >4.6 was a satisfactory predictor of patients’ survival after acute myocardial infarction [37]. Additional studies should be conducted to verify the association between the parameters included in the time-domain analysis and OSA.

We noted that BMI was positively associated with OSA; this finding is consistent with that of another study [38]. Studies have identified classical risk factors for OSA including age, male sex, obesity (BMI > 30), snoring, high blood pressure, metabolic syndrome, and sleep duration of ≥8 h [38,39,40,41]. The strength of this study was that AHI (an indicator of OSA severity) was significantly associated with higher log LF/HF power ratios (a surrogate of sympathetic tone) in the adult patients who underwent whole-night PSG, but it was not significantly associated with the parasympathetic tone, RMSSD. This current study has the following limitations. First, a small number of participants referred for PSG were included in this study (*n* = 62). Second, the recordings were performed during whole-night PSG. Thus, the sympathetic and parasympathetic tones might be different between the sleep and awake stages.

## 5. Conclusions

This study demonstrated that a higher log LF/HF power ratio, a frequency-domain parameter HRV, and BMI were positively and significantly associated with the AHI in the adult patients who underwent whole-night PSG. The higher log LF/HF power ratio indicated increased sympathetic nervous system activity. Frequency-domain HRV measure is a useful method for OSA screening.

## 6. Patents

Adults with OSA had associated with HRV, higher sympathetic tone and diminished vagal responsiveness [23]. In the current study, we suggested that a higher log LF/HF power ratio, higher sympathetic tone, might be better than vagal responsiveness for OSA prediction.

## Figures and Tables

**Figure 1 medicina-59-00471-f001:**
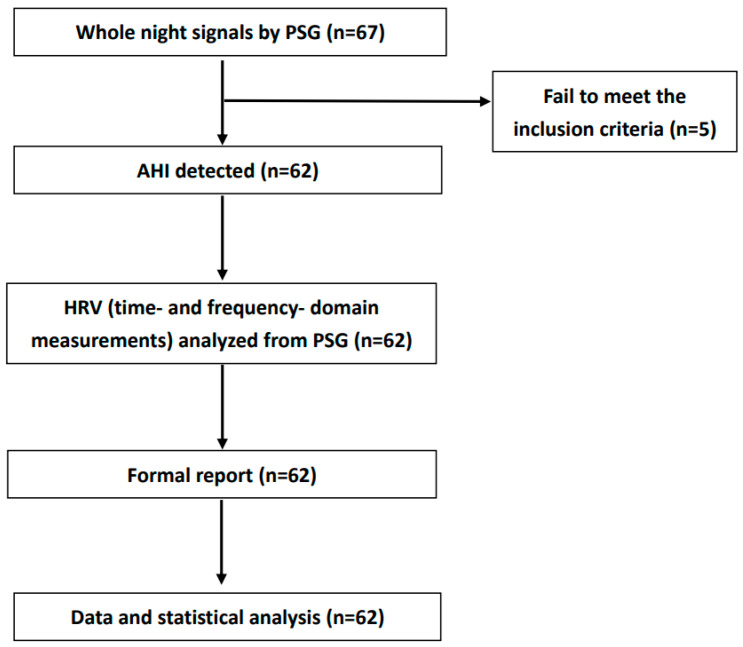
Block diagram of HRV data handling and analysis. Abbreviation: AHI: apnea–hypopnea index; PSG: polysomnography; HRV: heart rate variability.

**Figure 2 medicina-59-00471-f002:**
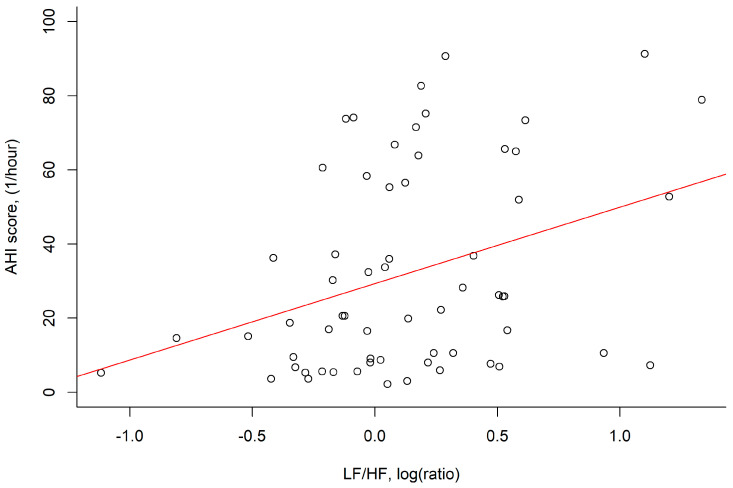
The LF/HF power, log (ratio) was positively correlated with the AHI score, (1/hour) in correlation analysis (*r* = 0.35, *p* = 0.006). Abbreviation: AHI: apnea–hypopnea index; HF: absolute power of the high–frequency band (0.015–0.4 Hz); LF: absolute power of the low–frequency band (0.04–0.15 Hz).

**Figure 3 medicina-59-00471-f003:**
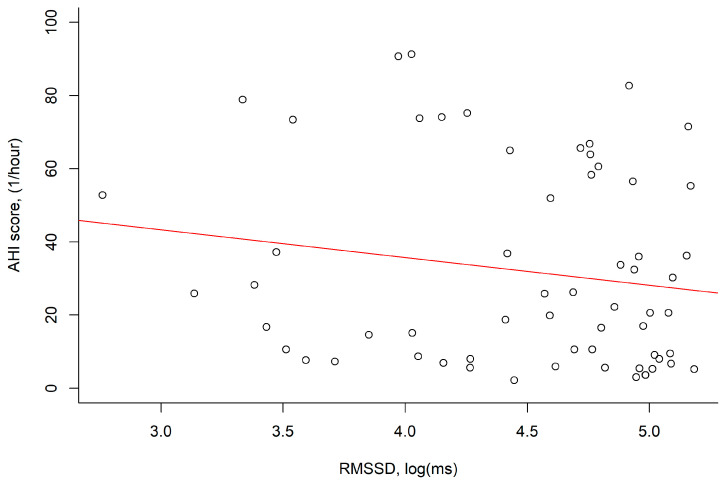
The RMSSD, log (ms) values were negatively correlated with the AHI score, (1/hour) (*r* = −0.17, *p* = 0.191). Abbreviation: AHI: apnea–hypopnea index; RMSSD: root mean square of successive differences.

**Table 1 medicina-59-00471-t001:** The demographic characteristics of the participants.

AHI Score	Total	AHI < 15	AHI ≥ 15	*p*
Gender				0.334
female	12	6	6	
male	50	17	33	
Age, year	44.4 ± 11.5	44.8 ± 12.0	44.2 ± 11.3	0.833
Recording time, hour	6.9 ± 0.3	6.9 ± 0.3	6.9 ± 0.3	0.897
BMI	28.7 ± 5.2	27.3 ± 4.2	30.0 ± 5.5	0.124
NC, cm	40.2 ± 4.8	38.5 ± 3.8	41.2 ± 5.1	0.016
AHI score (1/hour)	32.1 ± 27.0	7.1 ± 2.9	46.8 ± 23.9	<0.001
Hypoxemia index (1/hour)	9.7 ± 8.6	5.0 ± 2.5	12.5 ± 9.7	0.004
Arousal index (1/hour)	19.7 ± 13.9	10.4 ± 3.8	25.2 ± 14.9	<0.001
**Time domain**				
SDNN, log (ms)	4.8 ± 0.5	4.9 ± 0.5	4.8 ± 0.5	0.206
SDANN, log (ms)	4.2 ± 0.7	4.4 ± 0.7	4.1 ± 0.7	0.097
SDNNIDX, log (ms)	4.5 ± 0.5	4.5 ± 0.5	4.5 ± 0.6	0.761
pNN50, log (%)	3.1 ± 0.8	3.3 ± 0.8	3.0 ± 0.9	0.207
TINN, log (ms)	5.3 ± 0.4	5.3 ± 0.3	5.4 ± 0.5	0.443
SDSD, log (ms)	4.5 ± 0.6	4.6 ± 0.5	4.4 ± 0.6	0.303
RMSSD, log (ms)	4.5 ± 0.6	4.6 ± 0.5	4.4 ± 0.6	0.303
**Frequency domain**				
VLF, log (Hz)	6.3 ± 0.9	6.1 ± 0.7	6.4 ± 1.0	0.356
LF, log (Hz)	7.0 ± 1.2	7.1 ± 1.0	7.0 ± 1.3	0.856
HF, log (Hz)	7.0 ± 1.4	7.2 ± 1.1	6.9 ± 1.5	0.686
LF/HF, log (ratio)	0.1 ± 0.5	0.01 ± 0.5	0.2 ± 0.4	0.106

The distribution properties of quantitative variables were expressed using mean ± SD. Abbreviation: AHI: apnea–hypopnea index; BMI: body mass index; HF: absolute power of the high-frequency band (0.015–0.4 Hz); LF: absolute power of the low-frequency band (0.04–0.15 Hz); NC: neck circumference; pNN50: percentage of successive RR intervals that differ by more than 50 ms; RMSSD: root mean square of successive differences; SDANN: standard deviation of the average NN intervals for each 5 min segment of a 24 h HRV recording; SDNN: standard deviation of normal-to-normal (NN) intervals; SDNNIDX: mean of the standard deviations of all the NN intervals for each 5 min segment of a 24 h HRV recording; SDSD: standard deviation of successive RR interval differences; TINN: baseline width of the RR interval histogram; VLF: absolute power of the very-low-frequency band (0.0033–0.04 Hz).

**Table 2 medicina-59-00471-t002:** The correlation between the HRV and the demographic data.

Variables	Value	Age	BMI	NC	AHI	Hypoxemia Index	Arousal Index
**Time domain**							
SDNN, log (ms)	4.8 ± 0.5	0.01 (0.955)	−0.27 (0.036)	−0.31 (0.014)	−0.13 (0.308)	−0.23 (0.076)	<0.01 (0.997)
SDANN, log (ms)	4.2 ± 0.7	0.20 (0.115)	−0.20 (0.121)	−0.10 (0.450)	−0.09 (0.484)	−0.18 (0.158)	−0.05 (0.681)
SDNNIDX, log (ms)	4.5 ± 0.5	−0.10 (0.421)	−0.20 (0.113)	−0.30 (0.016)	−0.01 (0.945)	−0.19 (0.141)	0.07 (0.563)
pNN50, log (%)	3.1 ± 0.8	−0.09 (0.492)	−0.23 (0.072)	−0.28 (0.026)	−0.16 (0.204)	−0.30 (0.016)	0.05 (0.691)
TINN, log (ms)	5.3 ± 0.4	−0.37 (0.003)	0.27 (0.003)	−0.04 (0.740)	0.22 (0.090)	0.21 (0.096)	0.11 (0.384)
SDSD, log (ms)	4.5 ± 0.6	−0.08 (0.534)	−0.27 (0.034)	−0.38 (0.003)	−0.17 (0.191)	−0.25 (0.052)	−0.04 (0.782)
RMSSD, log (ms)	4.5 ± 0.6	−0.08 (0.534)	−0.27 (0.034)	−0.38 (0.003)	−0.17 (0.191)	−0.25 (0.052)	−0.04 (0.782)
**Frequency domain**							
VLF, log (Hz)	6.3 ± 0.9	−0.13 (0.311)	−0.02 (0.855)	−0.15 (0.233)	0.17 (0.197)	−0.03 (0.841)	0.19 (0.142)
LF, log (Hz)	7.0 ± 1.2	−0.11 (0.379)	−0.23 (0.070)	−0.35 (0.006)	−0.08 (0.524)	−0.17 (0.194)	0.03 (0.804)
HF, log (Hz)	7.0 ± 1.4	−0.15 (0.260)	−0.28 (0.030)	−0.42 (0.001)	−0.19 (0.139)	−0.27 (0.031)	−0.02 (0.903)
LF/HF, log (ratio)	0.1 ± 0.5	0.06 (0.624)	0.22 (0.080)	0.26 (0.040)	0.35 (0.006)	0.43 (0.001)	0.09 (0.473)

Data are presented in the mean ± SD fashion for value while in the *r* (*p* value) fashion for other variables. Abbreviation: AHI: apnea–hypopnea index; BMI: body mass index; HF: absolute power of the high-frequency band (0.015–0.4 Hz); LF: absolute power of the low-frequency band (0.04–0.15 Hz); NC: neck circumference; pNN50: percentage of successive RR intervals that differ by more than 50 ms; RMSSD: root mean square of successive differences; SDANN: standard deviation of the average NN intervals for each 5 min segment of a 24 h HRV recording; SDNN: standard deviation of normal-to-normal (NN) intervals; SDNNIDX: mean of the standard deviations of all the NN intervals for each 5 min segment of a 24 h HRV recording; SDSD: standard deviation of successive RR interval differences; TINN: baseline width of the RR interval histogram; VLF: absolute power of the very-low-frequency band (0.0033–0.04 Hz).

**Table 3 medicina-59-00471-t003:** The significant factors for AHI using the multiple linear regression.

	β	S.E.	t Value	*p*
BMI	2.20	0.59	3.72	<0.001
LF/HF, log (ratio)	15.01	6.71	2.24	0.029
Intercept	−33.08	17.04	−1.94	0.057

Abbreviation: BMI: body mass index; HF: high frequency; LF: low frequency; SE: standard error.

## Data Availability

The datasets generated during and/or analyzed during the current study are not publicly available, but are available from the corresponding author on reasonable request.

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
