# Peer review of "Association of Heart Rate Variability with Obstructive Sleep Apnea in Adults"

_medicina, 2023, doi:10.3390/medicina59030471_

Round 1

Reviewer 1 Report

This study included the polysomnographic results of 62 patients evaluated for OSA. Heart rate variability has been reported in OSA. The authors evaluated this variability using frequency and time domains. 

Major changes: the association of time and frequency domains with OSA was evaluated. However, the impact of hypoxemia and arousals was not included in the analysis. If unable to provide this data, the probability that arousals (and associated sympathetic activation) and not AHI could drive variability should be included in the discussion.

Minor changes: report the definition used for hypopneas in the AHI.

In table 1, please clarify how data is presented i.e. (mean+- SD)

Table 2, please clarify that p value is reported in parenthesis.

Author Response

Please see the attachment, thank you so much for your kind review.

Reviewer 2 Report

Dear Author(s)

A: There are grammatical errors that you should re-edit the text.

B: We consider AHI > 5 events/h for OSA cases. Why did you select a cutoff of 15 events/h?

C: Please add the full name of abbreviation below each table.

D: Please add the full name of each abbreviation for the first time in the abstract and the rest of the text such as LF/HF in the abstract and so on.

E: Please add the number of cases in Figure 1 for each section.

F: Please add the unit for each variable in the text, tables, and figures.

G: What are the data in Tables? r(P-value) or etc. Please write below the tables.

C:

Author Response

(The authors gave the same response as above.)

Round 2

Reviewer 1 Report

I thank the authors for the changes in their manuscript. 

I do not have any other comments. 

Reviewer 2 Report

Dear Author(s)

Thank you for your response. The manuscript is acceptable.